# Quantitative uniqueness of human brain evolution revealed through phylogenetic comparative analysis

Ian F Miller[1,2]*, Robert A Barton[3], Charles L Nunn[2,4]

[1]Ecology and Evolutionary Biology, Princeton University, Princeton, United States; [2]Department of Evolutionary Anthropology, Duke University, Durham, United States; [3]Evolutionary Anthropology Research Group, Department of Anthropology, University of Durham, Durham, United Kingdom; [4]Duke Global Health Institute, Duke University, Durham, United States

**Abstract** While the human brain is clearly large relative to body size, less is known about the timing of brain and brain component expansion within primates and the relative magnitude of volumetric increases. Using Bayesian phylogenetic comparative methods and data for both extant and fossil species, we identified that a distinct shift in brain-body scaling occurred as hominins diverged from other primates, and again as humans and Neanderthals diverged from other hominins. Within hominins, we detected a pattern of directional and accelerating evolution towards larger brains, consistent with a positive feedback process in the evolution of the human brain. Contrary to widespread assumptions, we found that the human neocortex is not exceptionally large relative to other brain structures. Instead, our analyses revealed a single increase in relative neocortex volume at the origin of haplorrhines, and an increase in relative cerebellar volume in apes.
DOI: https://doi.org/10.7554/eLife.41250.001

## Introduction

Primates vary almost a thousand-fold in endocranial volume – a measure which closely approximates brain size – ranging from 1.63 mL in mouse lemurs (*Isler et al., 2008*) to 1478 mL in humans (*Robson and Wood, 2008*). Body size is perhaps the most important statistical predictor of brain size across primates, with larger bodied species having larger brains, but substantial variation remains after accounting for the effects of body size (*Isler et al., 2008*). While numerous comparative studies have sought to identify ecological, behavioral, and cognitive correlates of this variability (*Barton, 1999*; *MacLean et al., 2014*; *DeCasien et al., 2017*; *Powell et al., 2017*; *Noonan et al., 2018*), much less is known about the evolutionary patterns and processes that generated extant variation in brain size within the primate clade, how these differ for different components of the brain, or the degree to which the brain phenotypes of particular species, such as humans, are the result of exceptional patterns of evolutionary change.

A common approach to investigating human uniqueness is to test whether humans fall 'significantly' far from a regression line, for example by regressing brain size on body mass (*Azevedo et al., 2009*; *de Sousa et al., 2010*; *Herculano-Houzel and Kaas, 2011*). One surprising recent result reported from such an analysis is that the mass of the human brain is only 10% greater than expected for a primate of human body mass (*Azevedo et al., 2009*). However, such non-phylogenetic methods may give misleading results because they fail to incorporate trait co-variation among species that results from shared evolutionary history. Valid analysis requires methods that account for phylogeny both when estimating scaling parameters and when evaluating deviations

*For correspondence:
ifmiller@princeton.edu

Competing interests: The authors declare that no competing interests exist.

**eLife digest** Humans have much larger brains than other primates, but it is not clear exactly when and how this difference emerged during evolution. Some scientists believe that the expansion of a part of the brain called the neocortex – which handles sight, hearing, conscious decision-making and language – drove the increase in the size of the human brain. Newer studies have challenged that idea.

One way to learn more about how humans evolved bigger brains is to compare the size of the brain, and specific parts of the brain, between humans and our closest relatives: non-human primates. To make accurate comparisons, scientists must account for many factors. Closely related primates may have more similar traits because they more recently shared a common ancestor. This means the evolutionary relationships between species need to be considered. Larger animals also tend to have larger brains so it is important to consider body size, too.

Now, Miller at al. show that the human brain is much larger than expected even after accounting for these factors, and that increases in brain size accelerated over the course of early human evolution. In the analyses, the brain and skull sizes of different living primate species, like chimpanzees and gorillas, and fossils of extinct primates, including Neanderthals, were compared using mathematical models.

These findings suggest that larger brains provided fitness advantages that led to large brain sizes in modern humans and Neanderthals. These increases in brain size were not driven by disproportionate growth in the neocortex alone, but rather by increases in the size of many parts of the brain. Increases in the relative size of the cerebellum, which is essential for balance and movement, were also important.

DOI: https://doi.org/10.7554/eLife.41250.002

from scaling patterns exhibited by individual species (*Garland and Ives, 2000*; *Ross et al., 2004*; *Organ et al., 2011*). An additional source of error arises if the species being investigated is included in the regression model (e.g. *Azevedo et al., 2009*), particularly when, as for humans, the phenotypic trait lies at the extreme of the distribution for the other species in the analysis. This procedure would reduce the magnitude of deviations from expected trait values for lineages that have undergone exceptional change, and in the case of humans, would bias the results toward failing to detect uniqueness.

Comparative methods make it possible to incorporate phylogeny into analyses and to model phenotypic evolution in ways that uncover hitherto hidden patterns. Such methods are now being applied to a wide variety of traits (e.g. *Vining and Nunn, 2016*; *Pagel, 1999*; *Orme, 2013*), including brain size. *Pagel (2002)* estimated phylogenetic scaling parameters to characterize the evolutionary trajectory of endocranial volume (ECV) in fossil hominins. His analyses revealed that ECV evolution accelerated towards the present. As this analysis did not account for body size, it is not clear to what extent this pattern reflects changes in brain size independent of body size. *Montgomery et al. (2010)* used ancestral state reconstruction with fossil data to demonstrate a directional trend in primate brain size evolution and to identify branches in the primate phylogeny along which exceptional evolutionary change occurred. They found that while the absolute change in the mass of the human brain was exceptional, the rate of change relative to body size was not. Phylogenetic methods have also been used to examine how specific brain components evolved and the extent to which the branch leading to humans exhibited unusual amounts or rates of change in the size of these components (*Barton and Venditti, 2013*; *Barton and Venditti, 2014*). Recently, *Lewitus (2018)* suggested that comparative analyses of neuroanatomical data can be improved by incorporating and comparing results from different evolutionary models.

Here, we use phylogenetic methods to model the evolution of brain size and to identify exceptional evolutionary change along phylogenetic branches. We employ three methods: The first method models trait evolution both as a multi-optima Ornstein-Uhlenbeck (OU) process (which incorporates stabilizing selection and drift) and as a Brownian motion process (*Felsenstein, 1985*), and then compares the fit of the two models. In cases where the OU model is favored, exceptional patterns of trait evolution are indicated by recent shifts in adaptive optima in humans' (or other

species') evolutionary lineage. In cases where the Brownian model is favored, we apply our second method, which is a phylogenetic outlier test that uses phylogenetic generalized least squares (PGLS) to predict a phenotype for a species and then compares observed and predicted values. With this method, we can assess whether humans are a phylogenetic 'outlier' relative to expectations based on their phylogenetic position and trait covariation in other primate species. Our last method tests for directional and accelerating evolution by fitting phylogenetic scaling parameters to data on deviation from trait expectations and evolutionary time, building on previous efforts with these approaches (*Pagel, 2002*).

Using the first two methods, we investigate the evolution of absolute brain size and brain size relative to body mass within primates. Absolute brain volume has been shown to predict cognitive ability in primates better than other metrics that account for body mass (*MacLean et al., 2014*; *Deaner et al., 2007*). However, brain size is highly correlated with body size (*Isler et al., 2008*), and as such it is difficult to interpret the significance of brain size alone. Additionally, accounting for body mass gives more insights into the significance of brain size in life history processes, as relative brain size better approximates relative investment in cognitive ability. Accounting for body mass is also important as the relationship between this trait and brain size is associated with scaling effects that reflect conservation of neural function, such as preservation of somatosensory acuity across large surface areas (*St Wecker and Farel, 1994*) and compensation for increased neural conduction distances in larger animals through (i) larger neuron and axon sizes, increased myelination, and increased white matter volume, all of which result in reduced neuron density (*Barton, 2012*; *Wang et al., 2008*; *Collins et al., 2013*) and (ii) increased neural resources devoted to prediction-based sensorimotor control that result from escalating neural conduction delays as body size increases (*More et al., 2010*). Other measures of relative brain size such as encephalization quotients, ratios, and residuals have been used in the past, but all make theoretical assumptions about the underlying relationship between brain and body size evolution that may not hold. Using relative measures can bias parameter estimates and is not recommended as a good statistical practice (*Freckleton, 2002*). Instead, an empirical approach is preferred in which the covariation of brain size with body size is accounted for within a statistical model that also accounts for phylogenetic history (such as PGLS).

We also apply the first two phylogenetic comparative methods to investigate the evolution of major brain structures involving the neocortex, cerebellum, and medulla. It is widely assumed that the neocortex expanded disproportionately relative to other brain structures during the evolution of anthropoid primates and most particularly in human evolution (*Kriegstein et al., 2006*; *Geschwind and Rakic, 2013*; *Florio and Huttner, 2014*). Surprisingly however, direct tests of this hypothesis are lacking, despite the focus of much evolutionary and developmental neuroscience on the neocortex as the site of interest for understanding human uniqueness and its developmental mechanisms (*Mitchell and Silver, 2017*). Recent evidence suggests that the cerebellum may have contributed more to human brain evolution than previously appreciated: it underwent rapid evolutionary expansion in the great ape clade including hominins (*Barton and Venditti, 2014*; *Smaers et al., 2018*) and has been implicated in shape changes of the brain in hominin fossil endocasts (*Kochiyama et al., 2018*; *Neubauer et al., 2018*). Molecular evidence now corroborates the proposal that selection on cerebellar function was an important feature of hominoid and hominin brain evolution (*Sousa et al., 2017*), with changes in protein-coding genes implicated in cerebellar development more likely to have evolved adaptively in apes than those implicated in neocortical development (*Harrison and Montgomery, 2017*). It therefore appears that the neocortex and cerebellum have had different evolutionary trajectories in primate evolutionary history. More research is needed to document and understand these patterns.

We examined volumetric change in the neocortex and cerebellum relative to both body mass and the volume of the rest of the brain. As a check to establish whether changes in evolutionary patterns for relative neocortex and cerebellum size are primarily attributable to changes in those structures or to changes in the rest of the brain, we investigated the evolution of the rest of the brain relative to body mass. We also conducted analyses of the volume of the medulla relative to body mass and the volume of the rest of the brain. The relative volume of the medulla does not vary significantly across clades (*Barton, 2000*) and as such it has not been attributed a major role in brain expansion. For the analyses of fossil species, brain component volumes are not available; thus, analyses of these lineages are restricted to overall brain size (ECV).

Although our main focus is on broad patterns across primate phylogeny and on the extent to which human brain evolution fits or departs from these patterns, we also examined brain evolution in other species that are considered to be unusually large-brained, such as the aye-aye (*Daubentonia*) and capuchins (*Cebinae*) (*Isler et al., 2008*; *Pagel and Harvey, 1989*). Our analyses also help to identify other primate species that have experienced exceptional expansion or reduction of the brain or its components, generating new questions for future research on exceptional brain evolution in primates.

We used our third method to characterize patterns of brain evolution in humans and extinct hominins. *Pagel (2002)* conducted similar analyses of raw ECV. Our analyses advance his findings in two ways. First, we incorporate body mass as a predictor. Second, we focus on the deviation from brain size expectations, based on the PGLS methods used to assess outlier status. Our findings therefore provide insights to the evolutionary trajectory of exceptional hominin ECV relative to primate-wide brain-body mass scaling relationships.

## Materials and methods

### Comparative data

We compiled ECV and female body mass data on non-human primates (*Isler et al., 2008*) as well as humans and fossil hominins (*Robson and Wood, 2008*, Tables *1* and *2*). Given that sex specific body mass estimates are available for ancient humans and extinct hominins (*Robson and Wood, 2008*), we used female values for body mass because female values are more tightly linked to ecological and life-history factors (*Gordon, 2006*) and sexual selection can drive increases in male body mass unlinked to ecology, obscuring brain-body scaling relationships (*Fitzpatrick et al., 2012*). We also compiled data on neocortex, cerebellum, and medulla volume (*Barton and Venditti, 2014*; *Stephan et al., 1981*; *Bush and Allman, 2004*). Values used to compute predictor variables (described below) for analyses of brain sub-structures were taken from *Isler et al. (2008)*. We used several phylogenies in our analyses. For analyses of hominin ECV, we constructed a 'hominin phylogeny' by combining the hominin consensus tree from *Organ et al. (2011)* and the non-human primate consensus tree from 10kTrees version 3 (*Arnold et al., 2010*). To ensure that our results in this set of analyses were not dependent upon the topology of the hominin phylogeny, we repeated them using an 'alternate hominin phylogeny,' constructed in a similar manner using another hominin tree from *Organ et al. (2011)*. Details of the tree construction process are given in Appendix 1. In all other analyses we used either the consensus primate phylogeny or a block of 100 primate phylogenies from 10kTrees, version 3.

To determine whether patterns of exceptional evolution represent absolute or relative changes in scaling, we included several predictor variables in our analyses. To investigate whether the volumes of structures changed relative to body size, we used body mass as a predictor variable, while we used a 'rest-of-brain' metric as a predictor variable to investigate whether the volumes of structures

**Table 1.** Hominin ECV and body mass data details.
All values are from *Robson and Wood (2008)*.

| Species | ECV (mL) | Sample size | Female body mass (kg) | Sample size |
|---|---|---|---|---|
| *Australopithecus africanus* | 464.00 | 8 | 30 | 7 |
| *Homo erectus* | 969.00 | 40 | 57 | 4 |
| *Homo habilis* | 609.00 | 6 | 32 | 2 |
| *Homo rudolfensis* | 726.00 | 3 | 51 | 2 |
| *Homo sapiens neanderthalensis* | 1426.00 | 23 | 65 | 7 |
| *Homo sapiens* | 1478.00 | 66 | 57 | 36 |
| *Paranthropus boisei* | 481.00 | 10 | 34 | 1 |
| *Paranthropus robustus* | 563.00 | 2 | 32 | 2 |
| *Australopithecus afarensis* | 458.00 | 6 | 30 | 4 |

DOI: https://doi.org/10.7554/eLife.41250.003

**Table 2.** Human brain data.

| Brain trait | Value | Source | Notes | Dataset |
|---|---|---|---|---|
| ECV | 1478.00 mL | (*Robson and Wood, 2008*) | Composite of values from 66 fossil specimens from locations across Eurasia and africa | 1 |
| Brain volume | 1267.65 mL | (*Barton and Harvey, 2000*) | Average of measurements of modern human brains | 2 |
| Brain volume | 1251.85 mL | (*Stephan et al., 1981*) | Measurement of modern human brain | 3 |

DOI: https://doi.org/10.7554/eLife.41250.004

changed relative to other brain structures. For the analyses of all structures other than the medulla, the 'rest-of-brain' was computed as whole brain volume – (neocortex volume +cerebellum vol). In analyses of the medulla, we calculated 'rest-of-brain' volume as brain volume - medulla volume. We also analyzed the volume of the 'rest-of-brain' [whole brain volume – (neocortex volume +cerebellum vol)] relative to body mass. The data sets used in all analyses, along with more detailed descriptions, are given in Appendix 1.

## Characterizing patterns of phenotypic evolution

We compared the fit of multi-optima Ornstein-Uhlenbeck (OU) models of evolution and Brownian models of evolution using a developmental version of the R package bayou (*Uyeda and Harmon, 2014*; *Uyeda, 2017*). OU models of evolution incorporate stabilizing selection and drift, while Brownian models only include drift. Bayou fits multi-optima OU models to a phylogeny using a Markov-Chain Monte Carlo (MCMC) approach. A shift in selection regime refers to a change in the parameters that determine the optimum trait value (towards which species evolve) at a specific location on a phylogeny. Thus, inferred changes in selective regime provide insights to how lineages differ. Shifts in selection regime along terminal branches of a tree would provide particularly strong evidence for a species' uniqueness.

*Grabowski et al. (2016)* proposed the following OU model to describe the evolution of a trait, *y*, as a function of a predictor variable, *x*:

*Equation 1*:

$$dy = -\alpha \left(y - y_0\right) dt + \sigma^2 dB$$

*Equation 2*:

$$y_0 = \theta + x\beta$$

In these equations, *dy* is the change in the trait value, $\alpha$ is the magnitude of the selective 'pull' towards the optimum trait value, $y_0$, and $\sigma^2$ is the variance of the white noise process *dB*. The variables $\theta$ and $\beta$ can be interpreted as the intercept and slope of the optimum regression line specified in *Equation 2*. The optimum regression line represents the state that a species is evolving towards rather than the actual evolutionary trajectory.

This model has limited utility when data for *x* are only available for the tips of the phylogeny because the values of *x* must be known along the branches of the phylogeny to infer the expected value of *y* for a lineage. We utilize two similar models implemented in the developmental version of bayou – the unweighted predictor model and the weighted predictor model (corresponding to 'immediate' and 'alphaweighted' options for 'slopechange' in bayou) – as these circumvent the issue of unknown phenotypes in ancestral lineages while incorporating a predictor variable into the OU model. The weighted predictor model considers the evolutionary history of the predictor variable while fitting models, and the unweighted predictor model only considers the values of the predictor variables at the tips of the phylogeny while fitting models. The details of these two models are provided in Appendix 2.

Bayou uses a MCMC to parameterize the models to fit the data by inferring the location and magnitude of concurrent shifts $\theta$ and $\beta$ on a phylogeny and by inferring the values of $\alpha$ and $\sigma^2$, which remain constant across the phylogeny. The parameters $\alpha$ and $\sigma^2$ are used in the calculation of the variance-covariance matrices used in evaluating model fit to the phylogeny. The phylogenetic half-life, the time needed for a trait to evolve halfway to the optimum, is computed as ln(2) / $\alpha$. We present phylogenetic half-life in units of tree height. A phylogenetic half-life less than tree height

indicates that the evolutionary processes can 'pull' parameter values to the optimum within the time-scale in question, while a phylogenetic half-life that exceeds tree height or constitutes a large percentage of tree height indicates that evolutionary processes have a weak 'pull' and trait values are not expected to closely approach the optimum during the timescale in question. The expected variance in trait values evolving to the same optima at equilibrium (stationary variance) can be computed as $\frac{\sigma^2}{2\alpha}$.

For each analysis, we ran the weighted and unweighted predictor models. We also ran a Brownian motion model in which the strength of stabilizing selection ($\alpha$) was fixed at $10^{-6}$ (resulting in a phylogenetic half-life ~9500 times greater than tree height; bayou cannot compute model likelihoods when $\alpha$ is 0), and no shifts away from the root regime were allowed. The predictor variable is still incorporated in the Brownian motion model, but no changes in its coefficient occur on the phylogeny. We used the hominin tree for the analysis of ECV and the consensus tree of extant primates for all other analyses. All MCMCs were run for 5,005,000 time steps, sampling every 10 time steps. The priors used are given in *Table 3*. For each analysis, two chains were run and checked for convergence in terms of likelihood, $\alpha$, and $\sigma^2$ (see Appendix 3 for discussion of chain non-convergence issues in analyses of ECV). We also checked for correlation in branch-wise posterior shift probability between chains. Diagnostic plots pertaining to chain convergence are given in *Source data 1*. The two chains were combined, with the first 30% of samples being discarded as burn in. We then obtained the likelihood of each model and calculated Bayes factors for each model pairing (*Kass and Raftery, 1995*; *Jeffreys, 1998*) using the *steppingstone* algorithm in bayou, which implements the method of *Fan et al. (2011)*. We imposed a posterior probability cutoff of 0.3 for shift detection.

When the multi-optima OU model was selected over the Brownian motion model, we used the location and magnitude of shifts in adaptive optima to assess changes in patterns of evolution. The inference of a shift on a terminal branch would indicate an exceptional pattern of evolution for a given species.

*Ho and Ané (2013)* identified several potential problems with OU models, including un-identifiability of parameters and over-fitting, but acknowledged that such models may be necessary, and recommended that Bayesian models, specifically bayou, be used to overcome these problems. Several other phylogenetic OU models have been developed (most notably *Hansen, 1997*), but none utilized Bayesian parameter estimation. *Cooper et al. (2016)* echoed the concerns of *Ho and Ané (2013)* and again recommended using Bayesian approaches. Additionally, they recommended weighing the fit of an OU model of evolution against that of a Brownian model, which do through our model selection process.

**Table 3.** Priors for bayou MCMC analyses.

| Model parameter | Prior distribution |
| --- | --- |
| $\alpha$ | Half-cauchy with scale factor 1. Fixed at 0 in Brownian model. |
| $\sigma^2$ | Half-cauchy with scale factor 0.1 |
| $\beta$ | Normal distribution with standard deviation = 0.5, mean = slope of linear model of trait and predictor data |
| $\theta$ | Normal distribution with standard deviation = 1, mean = intercept of linear model of trait and predictor data |
| Number of shifts per branch | Fixed at one |
| Branch-wise shift probability | Uniform |
| Number of shifts | Conditional Poisson distribution[*] with mean = 0.1*number of edges on phylogeny and maximum = number of edges on phylogeny. Fixed at 0 in Brownian model. |
| Location of shift along branch | Uniform |

[*]Calculated using 'cdpois' option in bayou.

DOI: https://doi.org/10.7554/eLife.41250.005

## Outlier detection using PGLS

When bayou indicated that the Brownian model of trait evolution was favored over the multi-optima OU model, we conducted a phylogenetic outlier test. This was accomplished using BayesModelS, an R script that generates distributions of predicted trait values for a species or several species based on phylogenetically controlled analyses of trait covariation with predictor variables (*Nunn and Zhu, 2014*). BayesModelS uses a Markov-Chain Monte Carlo (MCMC) to fit parameters of a PGLS model and assumes a Brownian motion model of evolutionary change. The PGLS models are used to generate trait value predictions for the species of interest. Uncertainty in phylogenetic structure can be accounted for by sampling from a set of trees (*Pagel, 2002*).

BayesModelS accounts for phylogenetic non-independence of residual trait values by incorporating branch scaling factors when fitting PGLS models. The MCMC samples between two branch length scaling factors, $\lambda$ and $\kappa$, to improve the fit of the models. The parameter $\lambda$ scales the internal branches of the phylogeny and measures phylogenetic signal (*Nunn, 2011*). Values for $\lambda$ were constrained to be in the interval [0, 1]. In the $\kappa$ model phylogenetic tree branch lengths are raised to the power $\kappa$. The value of $\kappa$ has previously been used to assess support for a 'speciational' mode of evolution (see *Pagel, 2002*).

When predicting the value of a trait for a species (or a group of species), its data were excluded from the BayesModelS analysis to avoid biasing the predictions. BayesModelS was then used to generate a posterior probability distribution of predicted values for that species, based on the predictor variable, estimated phylogenetic signal, and estimated trait co-variation with the other species in the analysis. Species were identified as outliers when their trait value was more extreme than 97.5% of the predicted trait values (i.e. when trait values fell outside 95% credible interval). A species was identified as a positive outlier when its true value fell above the majority of predictions, and a negative outlier when the opposite was true.

The analyses conducted using BayesModelS proceeded as follows. First, we investigated whether hominins follow primate brain size to body mass scaling rules by using BayesModelS to predict ECV based on body mass and phylogeny. We tested each hominin species for outlier status while excluding data on all hominins when generating predictions. When computing mean estimates for hominin ECV, we corrected for back transformation bias using the quasi-maximum likelihood estimator method described in *Smith (1993)*. We used the hominin phylogeny or the alternate hominin phylogeny in these analysis, and the data spanned 225 extant primate species (including humans) and 10 extinct hominin species.

Next, we identified individual primate species that are evolutionary outliers for ECV and other brain structures (neocortex, cerebellum, medulla, rest-of-brain). In these analyses, we accounted for phylogenetic uncertainty by using the block of 100 trees, which included *H. sapiens* and *H. neanderthalensis* but no other hominins. We iteratively tested each species in the data set for outlier status. Our analysis for ECV included data from 145 species, and our analyses for other brain structures structures included data from between 39 and 53 species.

MCMC chains were run for 1,000,000 time steps, and the first 200,000 time steps were discarded as burn in. Flat priors were used for all variables being predicted. To assess whether the post-burn in results were drawn from a stable distribution, we used the 'heidel.diag' function in the R package coda (*Plummer et al., 2006*). When post-burn-in results were not drawn from a stable distribution, we discarded an additional portion of the chain (as indicated by 'heidel-diag') so that only results drawn from a stable distribution remained. We ensured that the effective sample sizes for the PGLS model parameters (slope, intercept, most frequently selected phylogenetic scaling parameter) were greater than 1000 using the 'effectiveSize' function in coda (*Plummer et al., 2006*). Details of the MCMC diagnostics are given in supplementary materials S6, along with detailed results concerning the posterior predicted distribution and phylogenetic scaling parameters for each species in each analysis.

## Characterizing the tempo of ECV evolution in hominins

We investigated the evolutionary trajectory of brain-body scaling in hominins relative to other primates. We calculated the difference between observed ECV and the mean BayesModelS prediction for brain size (generated in the first described BayesModelS analysis in which data for all hominin species was excluded while generating predictions) for each of the hominin species. This difference,

which we call 'brain size deviation' represents the magnitude and direction of the deviation in brain size from what would be expected under primate brain-body scaling rules. We fit four PGLS model to hominin brain size deviation to examine how brain size deviation covaried with the phylogenetic distance from the hominin-*Pan* split: First, we fit a 'Brownian' model of brain size deviation with no predictor. We fixed λ at one in this and all subsequent models. Next, we fit a 'directional' model of brain size deviation predicted by phylogenetic distance from the hominin-*Pan* split, expecting to find a positive relationship between these variables if brain volume relative to body size has increased since the split of hominins and *Pan*. To determine whether evolutionary rates in brain size deviation have accelerated over time, we fit an 'acceleration' model that included the phylogenetic scaling parameter δ (*Pagel, 2002*; *Pagel, 1999*). Values of δ greater than one are consistent with accelerating evolution, but not necessarily directional evolution. Finally, we fit a 'directional acceleration' model in which we fit the parameter δ and used phylogenetic distance from the hominin-*Pan* split as a predictor of brain size deviation. In this model, a positive relationship between brain size deviation and phylogenetic distance, along with a value of δ greater than 1, would indicate that brain volume relative to body size has increased at an accelerating rate since the divergence of hominins from *Pan.* We compared these models using AICc. Analyses were conducted in the R package caper (*Orme, 2013*).

## Results

### Endocranial volume (ECV)

In the bayou analysis of ECV predicted by body mass using the hominin phylogeny, the Brownian model was favored over the weighted and unweighted predictor OU models with Bayes factors greater than 22. When we repeated this analysis using the alternate hominin phylogeny, we found that the un-weighted predictor OU model was favored over the weighted predictor OU model and the Brownian model with Bayes factors greater than 42, despite displaying poor convergence in terms of α and $\sigma^2$. However, both chains inferred a similar set of shifts, indicating that this is likely an issue related to parameter identifiability rather than to shift identifiability. In this model, progressive shifts towards larger ECV relative to body mass were detected within the hominin clade along the human lineage (*Figure 1A,B*). Shifts towards larger relative brain size were also detected on the terminal branch leading to *D. madagascariensis* and the internal branches leading to the *Lemuridae* and *Cebinae,* clades, and shifts towards smaller relative brain size were detected on the branch leading to the *Alouatta* clade, the branch leading to the clade containing the *Aotidae* and *Callitrichidae* families, and the branch leading to the *Colobinae* sub-family (*Figure 1—figure supplement 1*). The rejected weighted predictor OU model, as well as both OU models that were rejected in the bayou analysis using the hominin phylogeny, detected a very similar set of shifts that included shifts towards progressively larger ECV relative to body mass along the human lineage (*Source data 1*). Because the Brownian model was favored in the bayou analysis using the hominin phylogeny, we proceeded with BayesModels analyses using both the hominin and alternate hominin phylogenies.

In the BayesModelS analysis predicting ECV based on body mass while excluding all hominin data, the observed values for *H.* sapiens and *H. neanderthalensis* exceeded the mean values predicted by BayesModelS by 7.63 and 6.96 standard deviations respectively (*Figure 2C*). All hominin species were strongly supported positive outliers, with more than 99.9% of predictions falling below the observed values for ECV. The mean ECV prediction for a primate with the body mass of *H. sapiens* was 438 mL. Remarkably, the observed value for humans is 1478 mL, which is 238% greater than the mean of the predicted posterior distribution. A similar result was found for *H. neanderthalensis*; the observed ECV for this species exceeded the mean predicted value for a primate of their body mass by 952 mL, or 201%. Humans exceeded their predicted ECV by the greatest percentage, but all hominins exceeded predictions by at least 51% (*Figure 2C*, *Table 4*). We obtained similar results using the alternate hominin phylogeny (*Figure 2—figure supplement 1*, *Table 5*).

When we iteratively predicted ECV based on body mass and phylogeny for each species in the data set (no hominins besides *H. sapiens* and *H. neanderthalensis* were included in this analysis) and while using all data to generate predictions. We again found that humans were strongly supported positive outliers (Figure 4A). *H. neanderthalensis* was not identified as an outlier, perhaps because these analyses included all species except for the one being predicted, and thus inclusion of *H.*

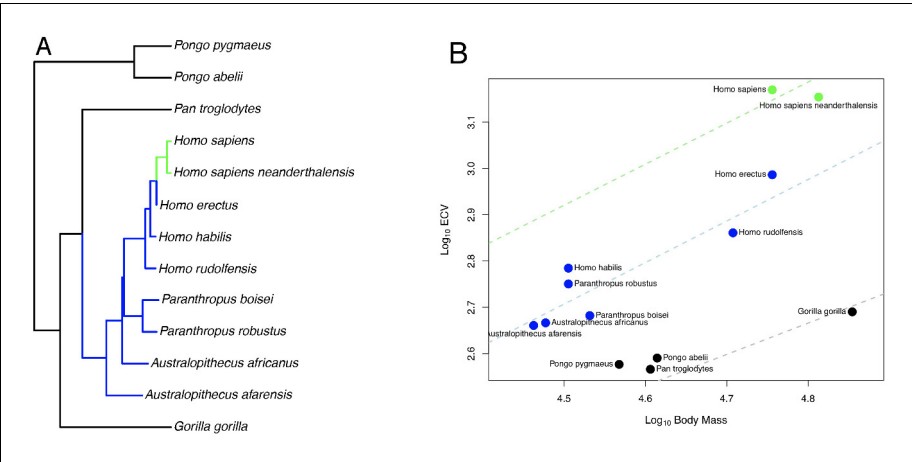

**Figure 1.** OU Model of ECV Evolution in Primates Panel. A shows the location of the selection regimes identified in an OU model of ECV predicted by body mass. Panel B shows the corresponding optimum regression lines representing the various selection regimes, along with body mass and ECV data. Data are colored by their corresponding selection regimes. All results are from the un-weighted predictor OU model in the bayou analysis using the alternate hominin phylogeny. Only the great ape clade is shown; selection regimes across the entire primate phylogeny are show in *Figure 1—figure supplement 1*.

DOI: https://doi.org/10.7554/eLife.41250.006

The following figure supplement is available for figure 1:

**Figure supplement 1.** OU Model of ECV Evolution in Primates Results are shown for the un-weighted predictor OU model of ECV predicted by body mass.

DOI: https://doi.org/10.7554/eLife.41250.007

*sapiens* resulted in a wide posterior distribution when predicting ECV in *H. neanderthalensis*. Indeed, when we excluded *H. sapiens* in this analysis we found that *H. neanderthalensis* was identified as a strongly supported positive outlier (*Source data 1*). We also identified several other primate species as outliers (see *Table 6* and *Source data 1*).

In the bayou analysis of ECV with no predictor variable using the hominin phylogeny, the Brownian model was selected over the un-weighted predictor OU models (in which the influence of the predictor was set to 0) with a Bayes factor >10. No weighted predictor model was run, as it would have been equivalent to the unweighted model given that no predictor variable was incorporated. An equivalent result was found when we repeated the analysis using the alternate hominin phylogeny. We then proceeded with the BayesModelS analysis, iteratively testing the outlier status of each species in the data set. We used the tree block for this analysis, and as such *H. sapiens* and *neanderthalensis* were the only hominins included. We found that neither humans nor Neanderthals were detected as an outlier (*figure 4—figure supplement 1*; *Source data 1*), indicating that without correcting for body mass, the variance in ECV across primates is great enough to prevent humans' brains from being detected as exceptionally large.

## Evolutionary trajectory of ECV in hominins

We conducted PGLS analyses of brain size deviation conducted to characterize the evolution of exceptional brain size in hominins (data shown in *Figure 3*). The analyses revealed evidence for both accelerated evolution of brain size deviation and directional evolution towards larger brain size deviations, as indicated by the directional acceleration model (AICc = −23.38) being favored over the acceleration (AICc = −21.93), directional (AICc = −17.56), and Brownian (AICc = −14.58) evolution models. In this best model, there was evidence of directional evolution towards larger brain size relative to body size (slope = 0.04) over time, and of accelerating evolution (δ = 8.36). These results suggest that the exceptionality of the human brain evolved recently. We found similar results when we repeated this analysis using the alternate hominin phylogeny (*Figure 3—figure supplement 1*). These analyses therefore support a model of accelerating evolution towards larger brain volume relative to body mass in *Homo sapiens*.

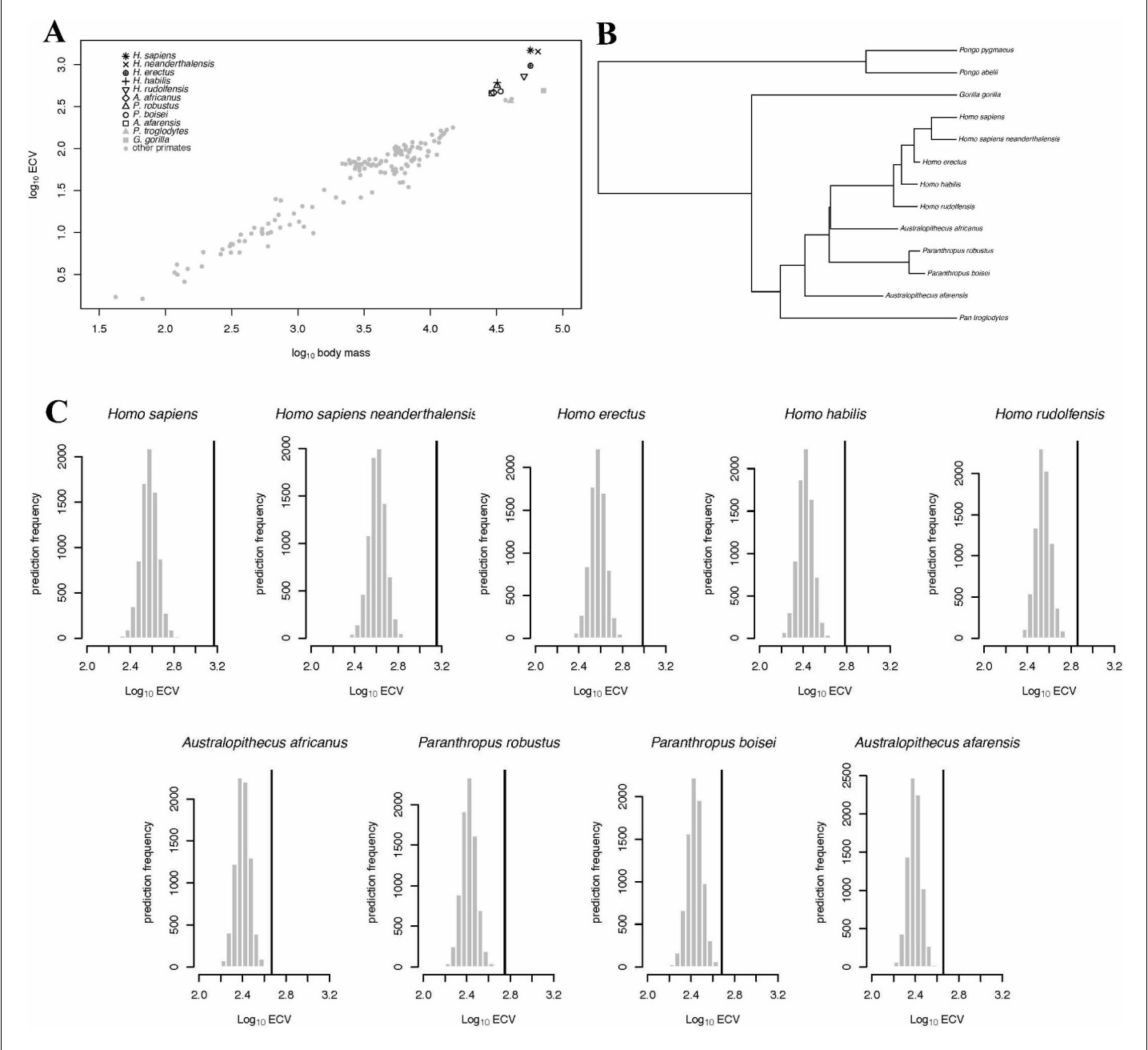

**Figure 2.** BayesModelS predictions of ECV in hominins. Panel (**A**) shows a scatter plot of primate ECV and body mass data. Panel (**B**) shows the topology of the great ape portion of the hominin phylogeny used in the BayesModelS analyses of hominin ECV. Panel (**C**) shows the posterior distributions of predicted ECV values generated by BayesModelS for hominin species with body mass used as the predictor variable. Vertical lines indicated observed values.

DOI: https://doi.org/10.7554/eLife.41250.008

The following figure supplement is available for figure 2:

**Figure supplement 1.** BayesModelS predictions of ECV in hominins.

DOI: https://doi.org/10.7554/eLife.41250.009

## Neocortex

In the bayou analysis of neocortex volume as predicted by body mass, the Brownian motion model was strongly favored over the weighted and unweighted predictor OU models, with Bayes

**Table 4.** Predicted Hominin ECV values from BayesModelS analysis using the hominin phylogeny.

| | True value (ml) | Corrected prediction (ml) | Difference (ml) | % difference |
|---|---|---|---|---|
| *Australopithecus africanus* | 464.00 | 294.73 | 169.27 | 57.43 |
| *Homo erectus* | 969.00 | 438.24 | 530.76 | 121.11 |
| *Homo habilis* | 609.00 | 306.83 | 302.17 | 98.48 |
| *Homo rudolfensis* | 726.00 | 409.63 | 316.37 | 77.23 |
| *Homo sapiens* | 1478.00 | 437.76 | 1040.24 | 237.63 |
| *Homo sapiens neanderthalensis* | 1426.00 | 474.46 | 951.54 | 200.55 |
| *Paranthropus boisei* | 481.00 | 319.00 | 162.00 | 50.78 |
| *Paranthropus robustus* | 563.00 | 307.60 | 255.40 | 83.03 |
| *Australopithecus afarensis* | 458.00 | 288.52 | 169.48 | 58.74 |

DOI: https://doi.org/10.7554/eLife.41250.010

factors > 18. Humans were detected as strongly supported positive outliers for neocortex volume by BayesModelS when body mass was used as the predictor variable (*Figure 4B*).

In the bayou analysis of neocortex volume with 'rest-of-brain' as the predictor variable, the weighted predictor model was selected over the unweighted predictor and Brownian motion models with Bayes Factors > 9.2. In the weighted predictor model, different scaling patterns were detected for strepsirrhines and haplorhines, with the optimum regression line for haplorhines falling above that of strepsirrhines. The only other detected transition in scaling occurred on the terminal branch leading to *Nasalis larvatus,* indicating a shift towards lower relative neocortex size (*Figure 5A,B*).

## Cerebellum

In the bayou analysis of cerebellar volume predicted by body mass, the Brownian motion model was favored over the weighted predictor and unweighted predictor OU models, with Bayes factors of 11.96 and 22.79, respectively. BayesModelS identified humans as strongly supported positive outliers for cerebellum volume when body mass was used as the predictor variable (*Figure 4C*).

In the bayou analysis of cerebellum volume relative to the rest-of-brain, the comparison between the unweighted predictor model and the Brownian motion model gave a Bayes factor of 10.65, while the comparison between the unweighted and weighted predictor models gave a Bayes factor of 0.20. This indicates that the OU models clearly outperform the Brownian model, but that neither OU model performs significantly better than the other. Both OU models detected a shift on the branch leading to apes associated with an increase in optimum cerebellar volume relative to the 'rest-of-brain' volume (*Figure 5C,D*).

**Table 5.** Predicted Hominin ECV values from BayesModelS analysis using the alternate hominin phylogeny.

| | True value (ml) | Corrected prediction (ml) | Difference (ml) | % difference |
|---|---|---|---|---|
| *Australopithecus africanus* | 464.00 | 288.18 | 175.82 | 61.00 |
| *Homo erectus* | 969.00 | 431.04 | 537.96 | 124.81 |
| *Homo habilis* | 609.00 | 300.16 | 308.84 | 102.89 |
| *Homo rudolfensis* | 726.00 | 401.94 | 324.06 | 80.62 |
| *Homo sapiens* | 1478.00 | 431.20 | 1046.80 | 242.76 |
| *Homo sapiens neanderthalensis* | 1426.00 | 468.41 | 957.59 | 204.44 |
| *Paranthropus boisei* | 481.00 | 311.41 | 169.59 | 54.46 |
| *Paranthropus robustus* | 563.00 | 299.74 | 263.26 | 87.83 |
| *Australopithecus afarensis* | 458.00 | 281.59 | 176.41 | 62.65 |

DOI: https://doi.org/10.7554/eLife.41250.011

**Table 6.** Summary of evidence for exceptional brain evolution among non-human primates.

| Species/Clade | Exceptional trait | Evidence |
|---|---|---|
| *Alouatta* | Reduced ECV relative to body mass | Shift in OU model |
| *Aotidae* and *Callitrichidae* | Reduced ECV relative to body mass | Shift in OU model |
| *Cacajao calvus* | Increased ECV relative to body mass | Outlier Detection |
| *Cebinae* | Increased ECV relative to body mass | Shift in OU model |
| *Cebus albifrons* | Increased cerebellum relative to body mass | Outlier detection |
| *Chiropotes satanas* | Reduced ECV relative to body mass | Outlier Detection |
| *Colobinae* | Reduced ECV relative to body mass | Shift in OU model |
| *Daubentonia madagascariensis* | Increased ECV relative to body mass | Shift in OU model |
| *Gorilla beringei*[*] | Reduced ECV relative to body mass | Outlier Detection |
| *Gorilla gorilla*[*] | Reduced neocortex relative to body mass | Outlier Detection |
| *Lemuridae* | Increased ECV relative to body mass | Shift in OU model |
| *Loris tardigradus* | Reduced medulla relative to the rest of brain | Outlier Detection |
| *Microcebus murinus* | Reduced medulla relative to the rest of brain | Outlier Detection |
| *Nasalis larvatus* | Reduced neocortex relative to the rest of the brain | Shift in OU model |
| *Otolemur crassicaudatus* | Reduced neocortex, cerebellum relative to body mass | Outlier Detection |
| *Pan troglodytes schweinfurthii* | Increased ECV relative to body mass | Outlier Detection |
| *Pan troglodytes troglodytes* | Reduced ECV relative to body mass | Outlier Detection |

[*]The dataset for this analysis did not contain any other gorilla species.

DOI: https://doi.org/10.7554/eLife.41250.012

## Medulla

In the bayou analysis of medulla volume predicted by body mass, the Brownian motion model was selected over the two OU models with Bayes factors > 7.4. BayesModelS identified humans as

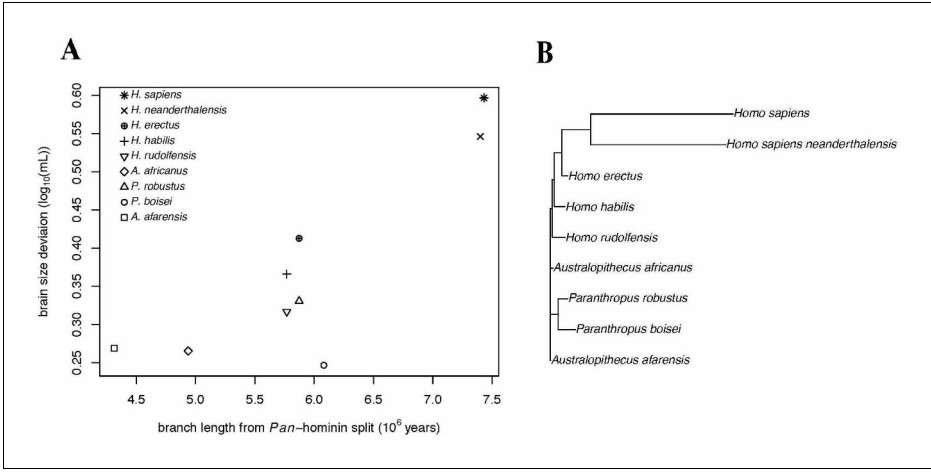

**Figure 3.** Accelerating Evolution of Brain Size Deviation in Hominins. (**A**) Brain size deviation was calculated as the difference between the mean BayesModelS prediction (made while excluding all hominin data from analysis and using the hominin phylogeny) and the observed value. Phylogenetic distance was measured as time since the shared ancestor of hominins and *Pan* at 7.43 mya. (**B**) Hominin clade in the hominin phylogeny after δ transformation, with δ = 8.36 following the directional acceleration model.

DOI: https://doi.org/10.7554/eLife.41250.013

The following figure supplement is available for figure 3:

**Figure supplement 1.** Accelerating Evolution of Brain Size Deviation in Hominins (alternate hominin phylogeny).

DOI: https://doi.org/10.7554/eLife.41250.014

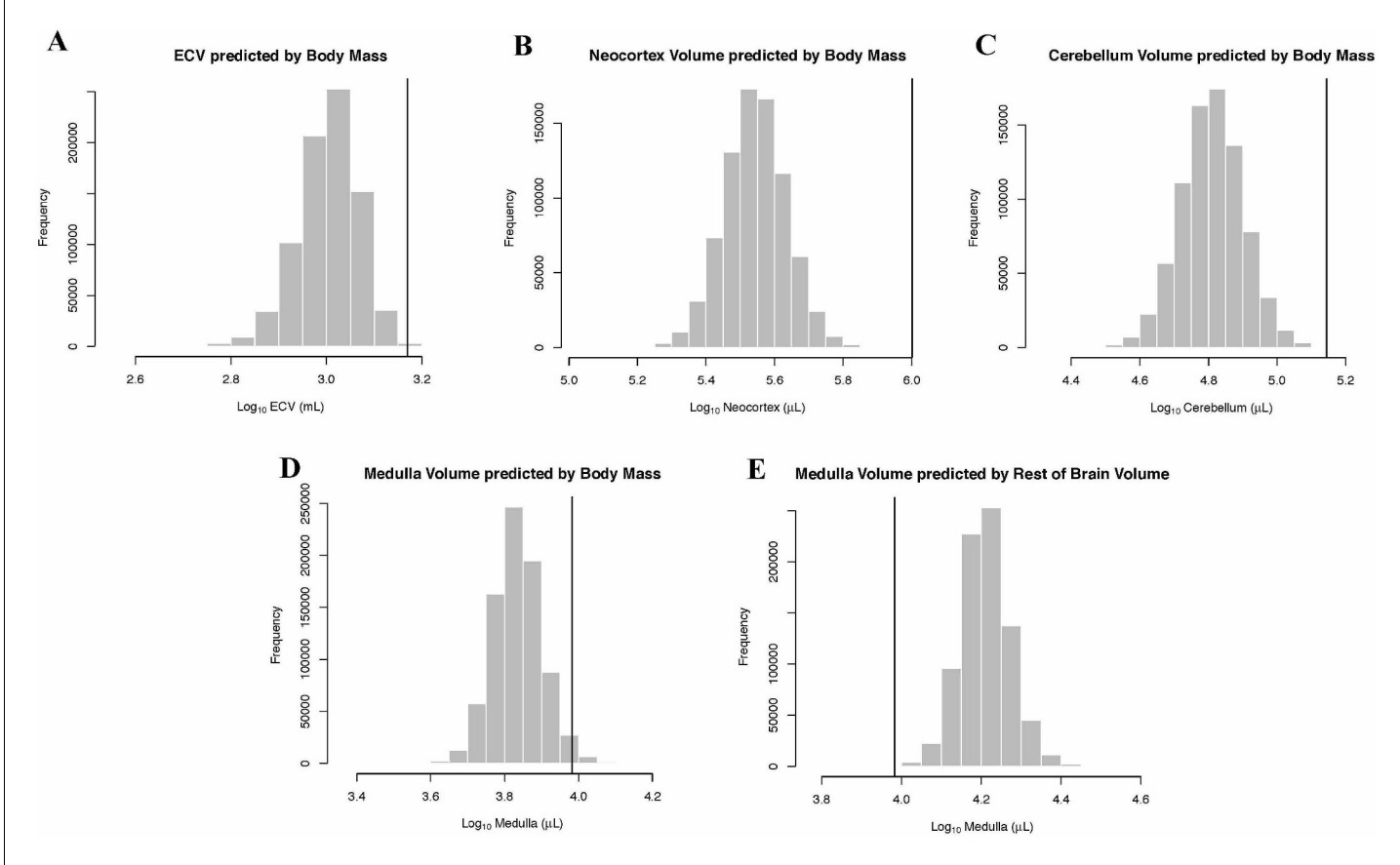

**Figure 4.** Human Outlier Status for Brain Traits Predicted distributions of trait values generated by BayesModelS are show as histograms. Vertical bars represent the observed values.

DOI: https://doi.org/10.7554/eLife.41250.015

The following figure supplement is available for figure 4:

**Figure supplement 1.** Human outlier status for ECV In the BayesModelS analysis of ECV with no predictor variable, humans were not detected as outliers.

DOI: https://doi.org/10.7554/eLife.41250.016

strongly supported positive outliers for medulla volume (*Figure 4D*). No other species were identified as exceptional in this analysis. When medulla was predicted by the 'rest-of-brain' volume, the Brownian motion model was again selected over the OU models, with Bayes factors > 3.8. Humans were identified as strongly supported negative outliers (*Figure 4E*).

### Rest-of-brain

In the bayou analyses of the rest-of-brain relative to body mass, the OU models were selected over the Brownian motion model, with Bayes factors > 13. However, the comparison between the two OU models gave a Bayes factor of 0.20, indicating that neither model is supported relative to the other. No shifts were detected in either model (*Figure 5E,F*).

## Discussion

Our phylogenetic analyses revealed that the human brain is 238% larger than the size expected for a primate of similar body mass and phylogenetic position. The exceptional size of the human brain was achieved through progressive scaling shifts towards larger size over several million years of hominin evolution, and the evolution towards increased brain size relative to expectations based on primate scaling patterns accelerated over time. These findings add an important dimension to previous

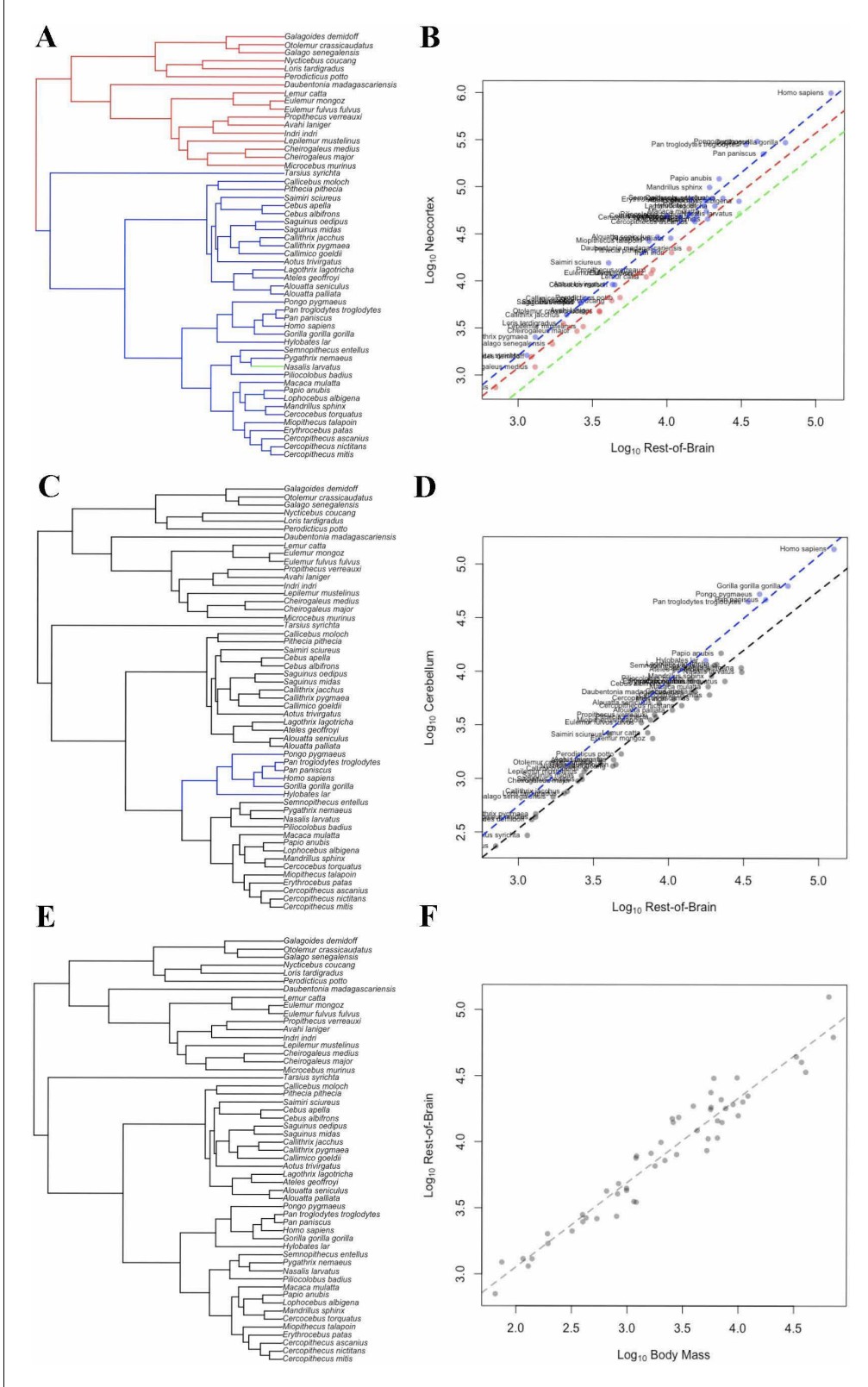

**Figure 5.** OU Models of Brain Structure Evolution in Primates. (**A and B**) correspond to the OU weighted predictor model of neocortex volume predicted by the rest-of-brain. (**C and D**) correspond to the OU unweighted predictor model of cerebellum volume predicted by the rest-of-brain. (**E and F**) correspond to the OU weighted predictor model of the rest-of-brain volume predicted by body mass. (**A, C**) and (**E**) show the location of

*Figure 5 continued on next page*

*Figure 5 continued*

selection regimes on the primate phylogeny. (B, D) and (F) show the optimum regression lines associated with the selection regimes. Points show primate trait and predictor data; colors correspond to the selection regimes. Colors in (A, C) and (E) match those in (B, D) and (F).

DOI: https://doi.org/10.7554/eLife.41250.017

observations of gradual phyletic increases in hominin brain size. *Du et al. (2018)* fit six evolutionary models to within- and between-lineage change in hominin brain sizes (random walk, gradualism, stasis, punctuated equilibrium, stasis-random walk and stasis-gradualism), obtaining the best fit for a gradualism model. However, their non-phylogenetic analysis did not test explicitly for accelerating directional increase. Our findings extend the results obtained by *Pagel (2002)* on absolute cranial volume, as the pattern of accelerating evolution is found even after accounting for body size. The pattern of accelerating brain size increase documented here is consistent with hypotheses that postulate a co-evolutionary positive feedback process driving human brain evolution, such as feedback between brain size and culture or language (*Wills, 1993*; *Deacon, 1998*) or between the brain sizes of conspecifics engaged in a socio-cognitive evolutionary arms race (*Dunbar, 1998*; *Miller, 2011*).

While humans clearly have the largest relative brain size among extant primates, anatomically modern humans were closely matched by *H. neanderthalensis*. However, even when accounting for the close phylogenetic relationship between humans and *H. neanderthalensis* and the exceptionally large brain of the latter, the human brain is still much larger than expected: humans were identified as strongly supported outliers when their ECV (relative to body mass) was predicted by phenotypic data from all primates, including *H. neanderthalensis*. This pattern was not reciprocal, however; *H. neanderthalensis* was not significantly different from other primates when *H. sapiens* was included in the model.

Significant variation exists between estimates of ECV and body mass made from different fossil specimens of the same hominin species (*Robson and Wood, 2008*). Thus, using single specimens to represent a species would not be a good statistical practice. We used a dataset in which almost all mean species values were calculated from multiple fossil specimens (*Table 1*). Unfortunately, we could not explicitly account for intraspecific variation in our analyses, as the multi-optima OU model fitting approach and the outlier test are unable to account for variation in both a trait and predictor variable. It would therefore be worthwhile to revisit our analyses as new phylogenetic comparative methods that can account for intraspecific variation become available. Additionally, data quality will likely improve over time. More hominin fossils will be discovered, increasing sample sizes for estimated ECV and body mass.

The hominin phylogeny will also likely become better resolved and more complete. We accounted for some phylogenetic uncertainty by repeating our analyses with an alternate phylogeny. The use of different phylogenies influenced outcomes of some statistical tests, as the Brownian model favored when we used the hominin phylogeny and OU model was favored when we used the alternate hominin phylogeny. However, we found that all of the OU models we fit inferred the same pattern of evolution towards larger ECV along the human lineage. The results of our outlier tests and PGLS model fitting – which assume a Brownian mode of evolution – also detected this pattern on different phylogenies. Collectively, these results indicate that our findings are likely to be robust to variations in assumed evolutionary relationships, and potentially to assumptions about the mode of evolution.

It is widely assumed that primate brain size evolution in general, and the large size of the human brain in particular, reflects expansion of the neocortex relative to other brain structures (*Kriegstein et al., 2006*; *Rakic, 2009*). Our results contradict this assumption: human neocortical volume was exceptionally large relative to body mass, but not exceptional relative the volume of the rest of the brain. We documented only one shift in neocortex size relative to the rest of the brain during primate evolution: an increase at the origin of all haplorrhines. This shift may be related to the visual specializations of haplorrhines for high-acuity photic vision, mediated by extensive cortical visual areas that make up over 50% of the cortex in these species (*Drury et al., 1996*; *Barton, 1998*; *Barton, 2007*). On branches postdating the split between haplorrhines and strepsirrhines, neocortex size is largely predictable from its scaling relationship to the rest of the brain, in line with the

proposed importance of cortical-subcortical connectivity in primate brain evolution (*Whiting and Barton, 2003*).

In contrast, we found that the cerebellum increased in size relative to the rest of the brain on the branch leading to apes. This finding is consistent with the results of recent studies implicating the cerebellum, and especially the lateral cerebellum, in brain expansion in apes and some other mammalian lineages (*Barton and Venditti, 2014*; *Smaers et al., 2018*; *MacLeod et al., 2003*). Our findings also reinforce the argument that subcortical structures should be given greater consideration in studies of mammalian brain evolution and cognition (*Barton, 2012*; *Miller and Clark, 2018*). Cerebellar specialization in apes may have been initiated by the demands on motor control and route-planning imposed by arboreal below-branch locomotion and/or by complex extractive foraging (*Barton and Venditti, 2014*; *Barton, 2012*). The fact that shifts in the relative size of neocortex and cerebellum occurred on different parts of the tree supports the theory of mosaic brain evolution (*Barton and Harvey, 2000*) and suggests that no single adaptive hypothesis is likely to be capable of accounting for primate brain evolution; rather, different selection pressures, on different information-processing capacities, likely operated at different times on different lineages.

Consistent with previous studies, we found that the medulla expanded in humans (positive outlier status for medulla volume relative to body mass), but to a lesser degree than other structures (negative outlier status for medulla volume relative to the rest of the brain). Relative to body mass, medulla volume has been shown to be much less variable across taxa than other brain structures, particularly compared to the neocortex and cerebellum. For example, unlike neocortex and cerebellum, medulla volume does not differ significantly between insectivores, strepsirrhines and haplorrhines (*Barton, 2000*). Accordingly, we found that after controlling for either body mass or brain size, the evolution of the medulla was not modulated by selection towards a stationary optimum in the primate clade. These results further support mosaic brain evolution (*Barton and Harvey, 2000*), and also suggest that scaling constraints related to connectivity with other brain regions (*Montgomery et al., 2016*) was less critical for the medulla than for the neocortex and cerebellum.

Several non-human primate species exhibited exceptional brain evolution in one trait or another, but only humans showed exceptional brain evolution for multiple brain components. As predicted, we detected shifts towards larger brain size on the terminal branches leading to *D. madagascariensis*, and on the branch leading to the *Cebinae* clade. Large brain size in *Daubentonia* and *Cebinae* has been attributed to extractive foraging and tool use (*Kaufman, 2005*; *Melin et al., 2014*; *Parker, 2015*). Although not one of our *a priori* expectations, we also documented shifts towards smaller brain size on branches leading to several clades, including *Alouatta*. We also found that two *Gorilla* species exhibit a smaller brain or neocortex size relative to body mass than expected. Given the extremely large body mass of *Gorilla* species, these unique traits may be the byproduct of a body mass increase rather than a reduction in brain size. Also unexpectedly, two *Pan troglodytes* sub-species were found to have exceptionally large and small ECV relative to body mass respectively. However, because more closely related species are weighed more heavily when BayesModelS generates distributions of predicted trait values, sister taxa deviating from expectations in opposite directions could result in both taxa being identified as outliers, even if they both conform to patterns of brain-body scaling for other primates. If the trait distributions for each species overlap significantly, then accounting for intraspecific variation in future analyses could remedy this problem.

The unexpected patterns that we observed amongst non-human primates raise several questions for further research. Given the well-established positive correlation between overall brain size and extended life history (*Isler and van Schaik, 2009*; *Sol, 2009*; *González-Lagos et al., 2010*), what are the life history implications of mosaic shifts in the sizes of different structures, and do these support any specific interpretations of the correlation between brain size and life histories? One hypothesis, the developmental costs hypothesis, is that large brains simply take longer to grow and mature, leading to extended periods of maternal investment and slower maturation, with other life history correlates of brain size being byproducts of developmental prolongation. Support for this hypothesis is provided by the finding that, amongst mammals, the durations of gestation and lactation have independent effects on pre- and postnatal brain growth, and once these effects are accounted for, other life history correlates are non-significant (*Barton and Capellini, 2011*). Despite their generally correlated evolution (*MacLeod et al., 2003*), we found shifts in the relative size of neocortex and cerebellum on different parts of the phylogenetic tree. Because these two structures have different developmental trajectories, the developmental costs hypothesis predicts different life

history correlates; this prediction has now received support (*Powell et al., 2019*). Further work is needed to establish exactly what developmental changes allowed for the neocortex and cerebellum to rest-of-brain scaling rules to change at the origin of haplorrhines and hominoids, respectively.

Another area of interest concerns the cases we found of brain or brain component size reduction. *Montgomery et al. (2010)* found that brain size reductions were rare during primate evolution, and that there was a general trend for brain size to increase across multiple branches of the phylogeny. This raises questions for future work concerning the causes, developmental mechanisms and functional implications of specific types of size reduction, such as those that we uncovered in brain size relative to body size in *Alouatta* and other clades, and in neocortex size relative to the rest of the brain in *N. larvatus*.

Finally, a key question that has attracted considerable attention concerns the ecological and social drivers of brain size and structure across large-scale evolutionary radiations. It has become increasingly apparent that correlations between overall brain size and behavioral ecology needed to be treated with caution (*Powell et al., 2017*; *Healy and Rowe, 2007*; *Wartel et al., 2018*). However, as suggested by the hypothesis of mosaic brain evolution, correlations between ecology and individual, less functionally heterogenous brain components may be more reliable and robust (*Barton and Venditti, 2014*; *Barton, 2012*; *Barton, 2007*; *Whiting and Barton, 2003*; *Montgomery et al., 2016*; *Barton et al., 1995*). Our analyses focused on gross subdivisions within the brain, and we suggest that further insights could be obtained by applying the phylogenetic methods used in this paper to more fine-grained neuro-anatomical data, using this approach to tease apart the contributions of correlated and mosaic change among brain components (*Melin et al., 2014*) and by incorporating ecological, behavioral, and developmental predictor variables that may account for additional variation in the traits of interest.

In conclusion, we provided robust evidence for directional and accelerating selection towards larger brain size over the course of human evolution, resulting in the human brain being exceptionally large for a primate of similar body mass. We also found that the sizes of human brain components – including the neocortex, cerebellum, and the rest of the brain – are not larger or smaller than expected relative to the size of the rest of the brain, but all are larger than expected for a primate of similar body mass. These results suggest that relative neocortical expansion is not a hallmark of our species. The diversity of evolutionary patterns for various brain components that we observed within primates suggests that no single factor fully explains primate brain evolution; instead, comparative research should investigate how different selection pressures influenced the evolution of different neuroanatomical components at different times on different parts of the phylogenetic tree. Additionally, future work should seek to analyze the evolution of other brain traits, including neuronal composition, using similar phylogenetic comparative methods that account for the non-independence of data from related species.

## Acknowledgments

We are grateful to Josef Uyeda for advice on using the developmental version of bayou. We would also like to thank Tom Milledge and Duke Research Computing for assistance with the Duke computing cluster.

## Additional information

### Funding

| Funder | Grant reference number | Author |
| --- | --- | --- |
| National Science Foundation | BCS-1355902 | Charles L Nunn |

The funders had no role in study design, data collection and interpretation, or the decision to submit the work for publication.

### Author contributions

Ian F Miller, Software, Formal analysis, Investigation, Visualization, Methodology, Writing—original draft, Writing—review and editing; Robert A Barton, Conceptualization, Data curation, Investigation,

Writing—original draft, Writing—review and editing; Charles L Nunn, Conceptualization, Supervision, Funding acquisition, Investigation, Methodology, Writing—original draft, Writing—review and editing

### Author ORCIDs
Ian F Miller    http://orcid.org/0000-0002-2673-9618
Charles L Nunn    http://orcid.org/0000-0001-9330-2873

### Decision letter and Author response
Decision letter https://doi.org/10.7554/eLife.41250.026
Author response https://doi.org/10.7554/eLife.41250.027

## Additional files
### Supplementary files
• Source code 1. Representative Code. Representative R code files for the bayou analyses ('representative bayou code.R'), BayesModelS analyses ('representative BayesModels code.R'), and pgls model fitting ('pgls models.R'), are contained in the this file, along with the BayesModelS code ('mult.spec.BayesModelS_v24.R') and other necessary data files.
DOI: https://doi.org/10.7554/eLife.41250.018

• Source data 1. Bayou and BayesModelS Results Details. Bayou Results details: Diagnostic plots giving details of chain convergence are provided in the 'bayou results summary.html' file along with detailed information on all OU and Brownain motion models for each trait and predictor pair. BayesModelS Results Details: Details of the BayesModelS results and diagnostic parameters of MCMC chains are given in the 'BayesModelS.results.csv' and 'BayesModelS.results.hominins.removed.csv' files.
DOI: https://doi.org/10.7554/eLife.41250.019

• Source data 2. All data and trees used in our analyses. Contains the following files: 1. data set 1. csv 2. data set 2.csv 3. data set 3.csv 4. consensus.tree.txt 5. tree.block.txt 6. grafted.tree.txt
DOI: https://doi.org/10.7554/eLife.41250.020

• Transparent reporting form
DOI: https://doi.org/10.7554/eLife.41250.021

### Data availability
All data used in our analyses are provided as supplementary material.

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

# Appendix 1

DOI: https://doi.org/10.7554/eLife.41250.022

## Data Compilation

All data and trees used in our analyses are included in the *Source data 2* file. We compiled three data sets for our analyses. The first was used for the analyses of endocranial volume (ECV), the second was used for the analyses of the neocortex and cerebellum, and the third was used for the analyses of the medulla.

In the first data set ('data set 1.csv'), we compiled ECV and female body mass values for non-human primates from *Isler et al. (2008)*, who compiled their data set in part from *Araújo et al. (2000)*, *Gordon, 2006*, *Smith and Jungers (1997)*, and *Thalmann and Geissmann (2000)*. This dataset was supplemented with fossil data for ancient humans and extinct hominins from *Robson and Wood (2008)*. These authors provided two taxonomies: one that recognized more species of hominins (the 'splitting taxonomy'), and another that lumped hominin lineages into fewer taxonomic categories (the 'lumping taxonomy'). We extracted values from the splitting taxonomy, except those for *Australopithecus africanus*, which were only available from the lumping taxonomy. We also chose to use values for *Homo erectus (sensu lato)* from the lumping taxonomy, as these values was calculated from fossils attributed to both *H. erectus* and *H. ergaster*, two species that are not differentiated in our phylogeny. We did not include *H. heidelbergensis* in our analyses because its phylogenetic position is unresolved (*Mounier and Caparros, 2015*). Sample sizes are given in *Table 1*. Museum numbers for the specimens used in calculating species mean values are given in Appendix I of *Robson and Wood (2008)*.

In the second data set ('data set 2.csv'), body mass, neocortical volume, and cerebellar volume for humans and extant non-human primates were compiled from the data set of Barton and Venditti (*Barton and Venditti, 2014*). We also complied brain volumes to use in the calculation of the 'rest-of-brain' predictor trait in the analyses of these brain structures. These values were calculated as an average of the values given in *Stephan et al. (1981)*, *MacLeod et al. (2003)*, *Bush and Allman (2003)*, *Rilling and Insel (1998)*, and *Rilling and Insel (1999)*. The second data set was limited to extant primates and included values for 55 species, including humans.

The third data set ('data set 3.csv') included body mass, brain volumes, and medulla volumes from *Stephan et al. (1981)*. This data set spanned 41 species.

A summary of all human ECV and body mass estimates and the analyses in which they were used is given in *Tables 1* and *2*.

All trait and predictor values were $\log_{10}$ transformed prior to analyses. When differences between component volumes were used in analyses, we calculated the logarithms after subtraction.

We used several different phylogenetic trees and tree blocks in our analyses. We constructed a 'hominin phylogeny' ('hominin.phylogeny.txt') that included humans, extinct hominins, and extant primates for use in the analyses of hominin ECV (including the analyses of directional and accelerating evolution); this phylogeny was produced by grafting the 'combined dataset consensus time tree' of hominin evolution from *Organ et al. (2011)* onto the time-scaled consensus tree of extant primates from version 3 of 10kTrees (*Arnold et al., 2010*). We grafted the clade (including the root branch) containing *Pan* and all fossil hominins onto the node at which *Gorilla* diverged from the *Pan* lineage, and then re-scaled this pasted clade so that the human tip lined up with those of extant primates. We also constructed an 'alternative hominin phylogeny' ('alt.hominin.phylogney.txt' using the 'morphology and molecular graft time tree' from *Organ et al. (2011)*. To construct this tree, we again grafted the clade (including the root branch) containing *Pan* and all fossil hominins onto the node at which *Gorilla* diverged from the *Pan* lineage, and then shortened the root branch so that the human tip lined up with those of extant primates. We were not able to use this method for constructing the hominin phylogeny because it would have resulted in the branch leading to

the clade containing *Pan* and hominins having a negative length. Both hominin phylogenies we constructed include humans, 300 other extant primates, and 13 extinct hominin species. In other analyses, we used a consensus tree ('consensus.tree.txt') of extant primates (for OU model fitting) or a block of 100 primate trees ('tree.block.txt') downloaded from version 3 of 10kTrees (*Arnold et al., 2010* for phylogenetic prediction).

## Appendix 2

DOI: https://doi.org/10.7554/eLife.41250.022

### Details of Bayou models

The un-weighted predictor model is described by the following equation:

*Equation 1*

$$E[y] = W\theta_M + x\,\beta_n$$

E[y] is the expected value of a species trait. W and $\theta_M$ represent the evolutionary weight matrix and $\theta$ matrix described in **Butler and King (2004)**. W is a 1 x *n* matrix whose entries are the weights given to each of the n selection regimes through which the species of interest evolved. The weight of each regime is dependent upon the phylogeny and the value of $\alpha$. More recent regimes have greater weights, especially when $\alpha$ is high. $\theta_M$ is an *n* x 1 matrix of the θ values of the regimes through which the species of interest evolved. The product of W and $\theta_M$ gives the effective $\theta$ value for a species that evolved towards the various optimum $\theta$ values specified in $\theta_M$. $\beta_n$ is the $\beta$ value of the parameter regime at the tip of the phylogeny. Therefore, in this model, the expected phenotype for a species is a function of the evolutionarily weighted effective $\theta$ value, the coefficient of the predictor variable of the current selection regime, and the value of the predictor at the tip of the phylogeny.

The weighted predictor model is described by a similar equation:

*Equation 2*

$$E[y] = W\theta_M + x\,W\beta_M$$

$\beta_M$ is an *n* x 1 matrix of the optimum $\beta$ values $\theta_M$. of the n regimes through which the species evolved, and is analogous to $\theta$ Thus, in this model the expected trait value of each species is a function of the species evolutionarily weighted effective $\beta$ and $\beta$ values, and the value of the predictor variable x at the tip of the phylogeny.

## Appendix 3

DOI: https://doi.org/10.7554/eLife.41250.022

### Problems with MCMC convergence in bayou

Bayou returned several MCMC chains during the analyses of ECV that did not converge in terms of likelihood, $\alpha$, and $\sigma^2$. To address this issue, we generated up to six MCMC chains in each analysis for both for the un-weighted predictor, weighted predictor, and Brownian models. Several chains with exceptionally high mean likelihood had $\sigma^2$ values approaching zero and very high $\alpha$ values that appeared to be bounded by a maximum value. We infer from these patterns that the chains were settling on an unrealistic pattern of evolution with the stationary variance approaching zero. These chains also inferred shifts erratically; they predicted shifts with posterior probability greater than 0.1 on many branches, but no shifts had a posterior probability greater than 0.3. We discarded these chains, and then selected the two chains with the highest mean likelihood for each analysis for subsequent use.

