## [Decision Letter]

Thank you for submitting your article "Quantitative uniqueness of human brain evolution revealed through phylogenetic comparative analysis" for consideration by *eLife*. Your article has been reviewed by four peer reviewers, and the evaluation has been overseen by Jessica Thompson as the Reviewing Editor and Patricia Wittkopp as the Senior Editor. The following individuals involved in review of your submission have agreed to reveal their identity: Christopher Heesy (Reviewer #2); Theodore Stankowich (Reviewer #3); Dietrich Stout (Reviewer #4).

These reviewers were all in general agreement that this is an important topic and the manuscript should be published after some revisions. To add strength to this, all came from different perspectives (paleoanthropology, neuroanatomy, phylogenetics), and all were careful not to overstep the boundaries of their individual knowledge. The reviewers have discussed the reviews with one another and the Reviewing Editor has drafted this decision to help you prepare a revised submission.

Summary:

This is a well-written manuscript that addresses an important issue in hominin evolutionary studies. The authors provide an excellent and up-to-date summary of current research, and identify clear gaps in information that need to be addressed through future research. Their results are novel, interesting, and will certainly stimulate much discussion. The authors apply a newer Bayesian modeling approach to the pattern of primate brain volume evolution for the purpose of evaluating brain volume evolution in living and fossil humans. Their analyses that demonstrate differential rates and directions of brain volume evolution among primates and within certain clades are novel because they are shown within a single analysis, and not piecemeal across various studies. The comparisons between varying assumption sets and data sets are carefully handled and provide illuminating results.

This is clearly an important contribution to an important topic, and the impact statement conveys this. The topic of human brain expansion has been of enduring interest for a long time, and the particular contribution is very timely in light of recent papers (cited by the authors) that are either contradicted or meaningfully extended by the work presented.

The essential revisions are consolidated below. In general, these coalesce into the following recommendation: the authors should de-emphasize some of the implications of their results, publish a more comprehensive description of their sample set, and re-run some analyses with different sample subsets in order to explore some of the patterns that seem to have manifested in the first round of analysis. There should also be some minor expansion of discussion of some key points that were uncovered through their analysis, and give a bit more time to discussion of the literature and issues detailed in the comments below.

Essential revisions:

Analytical Revisions:

1) What was not addressed by the current study was the relevance to the Herculano-Houzel lab's analyses. The authors bring up objections to the methods (and, obliquely, the conclusions) of studies on scaling of neuronal and non-neuronal cells numbers in primate CNS, but do not conduct comparable analyses. The authors use volumetric data, but the relationship between CNS volume and cell number, composition, size, variability, and density are not directly comparable. I see two solutions. The authors could just tone down or eliminate their discussion of the cell number and density studies or, and this would be my preference, they could also re-analyze the neuronal + non-neuronal cell data using modern phylogenetic approaches. Whichever the authors decide, I still believe that this paper should be published.

2) Subsection “Endocranial volume (ECV)”: If the exceptionally strong effects of H. sapiens "washed out" any potential effect on H. neanderthalensis and probably many other species, why not rerun these tests excluding H. sapiens when testing H. neanderthalensis and both of these species when testing the other primates? It may allow detection of more subtle changes in brain evolution that are smaller in effect, but equally interesting, and greater patterns may emerge.

Dataset revisions:

1) It is good to hold sex constant, but of course sex cannot be known for most extinct hominins. Therefore, it seems that averages of both males and females (brain and body sizes) would be more appropriate comparisons in analyses where fossil hominins are included. If the authors choose not to do this, they should justify it.

2) Although there is some clarity about the phylogenetic relationships of extant primates, there is less consensus about the relationships of fossil hominins. The authors use a consensus tree of their own making, but it is not clear how the analysis would change if small details of this tree were shifted around. It would be extremely helpful if at least one alternative phylogeny was demonstrated to have little impact on the results. Otherwise, a minor shift in the phylogenetic status of a fossil taxon could invalidate the big take-home from this study.

3) It would be very useful to include in the supplementary information a list of which specimens (museum numbers) of which fossil hominins went into the figure in each of the datasets. Sample sizes are so small for many of these, that a single cranium can make a big difference in this computation. The use and quality of different data sets has been a huge issue for this field, which has been bedeviled by small sample sizes, difficulties in dealing with intraspecific variation, and the use of estimated metrics (ECV, body mass) that may be derived in different ways esp. across fossil and extant species. Both ECV and body mass estimates are notoriously variable in hominin paleontology even for individual fossils, let alone in terms of a "species norm". This is a general issue that really needs to be discussed for the whole data set, but I would point to one particular point of concern, which would be the source of the Homo sapiens data. From which population or populations, fossil or modern, does this derive? Implications esp. for body size could potentially be a factor in the different modeling results for H.s. vs. Neanderthals, just as the authors point out that unique traits of Gorilla may be a "byproduct" of body size change rather than brain change.

4) It is not clear why only a single non-African species – H. neanderthalensis – was included in this analysis, when there are many examples of ECV for H. erectus and H. heidelbergensis. If it is to restrict the analysis to evolutionary trajectories that could feasibly have led to modern humans, that is acceptable – but the authors should explicitly explain these decisions.

5) Similarly, I do not understand why African H. heidelbergensis was not included in this analysis. Is this because its phylogenetic position is poorly-understood, and the authors were restricting themselves to African specimens where there is still debate over the number of Middle Pleistocene species of Homo? At the very least, this omission should be justified. It is a critical link between H. ergaster and H. sapiens.

Discussion Revisions:

1) The current level of discussion of the results on medulla size evolution should be more explicit about their medulla analyses or just consider de-emphasizing these results. This is because predictions about why medulla volume might vary among primates is elusive. Either the authors should be more explicit about their medulla analyses or just consider de-emphasizing these results.

2) Subsection “Endocranial volume (ECV)”: If the weak OU model provides a better fit and suggests that all primates experienced similar evolutionary trajectories, how does that mesh with the previous result that they show variation among taxa in evolutionary rate? More explanation and interpretation of this result seems warranted.

3) Results of the non-human primates should be more thorough. Given the volume of interesting results, particularly in Table 2, spending more time probing and hypothesizing about individual changes is warranted. Analysis without interpretation and suggestion for future work leaves this section flat. There is some discussion of this here but given the support in the dataset, outlining some potential new specific hypotheses for how primate life history influences brain evolution would make this a much more powerful paper.

4) There are of course many different ways to characterize the "size" of the human brain: absolute, relative to body, relative to other brain part, relative to range size, relative to phylogeny, etc. As it stands the discussion seems to imply that our task is simply to select and apply the one correct way, but I suspect/hope the authors would agree that the most appropriate method depends on the question being asked. For example, many studies of comparative cognition suggest we should actually focus on absolute rather than relative brain size whereas the choice to control for body size or phylogeny is also related to certain assumptions and interests. These authors would be very well suited to provide a definitive discussion of this issue that would be a useful touchstone for the field, and would also more clearly articulate the specifically evolutionary (e.g. as opposed to functional) questions their method is optimized to address. Some of this is indirectly reflected in the discussion of the unexpected results for the chimpanzee sub-species and it would be nice to see a more comprehensive treatment.

---

## [Author Response]

Essential revisions:Analytical Revisions:1) What was not addressed by the current study was the relevance to the Herculano-Houzel lab's analyses. The authors bring up objections to the methods (and, obliquely, the conclusions) of studies on scaling of neuronal and non-neuronal cells numbers in primate CNS, but do not conduct comparable analyses. The authors use volumetric data, but the relationship between CNS volume and cell number, composition, size, variability, and density are not directly comparable. I see two solutions. The authors could just tone down or eliminate their discussion of the cell number and density studies or, and this would be my preference, they could also re-analyze the neuronal + non-neuronal cell data using modern phylogenetic approaches. Whichever the authors decide, I still believe that this paper should be published.

We clarify that the Azevedo et al., (2009) result we discuss is that the mass of the human brain is only 10% larger than expected (Introduction):

“One surprising recent result reported from such an analysis is that the mass of the human brain is only 10% greater than expected for a primate of human body mass [8].”

We also removed our reference to these results from the Abstract, and instead focus on the pattern of scaling shifts that we detected.

While our findings are not directly comparable to the Azevedo et al., (2009) analysis (which is from the Herculano-Houzel lab), they do illustrate how flawed methods can make the human brain appear less exceptional. We make no other mention of findings from Dr. Herculano-Houzel’s analyses in our manuscript. Analyzing the neuronal composition data with our methods would be very interesting, but it is beyond the scope of this paper, and the sample sizes are not yet sufficient for the methods that we employ. We added text to the end of the Discussion section indicating that this would be a useful and interesting future application of our methods:

“Additionally, future work should seek to analyze the evolution of other brain traits, including neuronal composition, using similar phylogenetic comparative methods that account for the non-independence of data from related species.”

2) Subsection “Endocranial volume (ECV)”: If the exceptionally strong effects of H. sapiens "washed out" any potential effect on H. neanderthalensis and probably many other species, why not rerun these tests excluding H. sapiens when testing H. neanderthalensis and both of these species when testing the other primates? It may allow detection of more subtle changes in brain evolution that are smaller in effect, but equally interesting, and greater patterns may emerge.

We re-ran this analysis while excluding data for *H. sapiens*. We did indeed find that *H. neanderthalensis* was identified as an outlier. The analyses were otherwise congruent; i.e., no other species was identified as an outlier in this new analysis as compared to the original analysis. We added text describing this result (Results section):

“Indeed, when we excluded *H. sapiens* in this analysis we found that *H. neanderthalensis* was identified as a strongly supported positive outlier (supplementary materials S6).”

Dataset revisions:1) It is good to hold sex constant, but of course sex cannot be known for most extinct hominins. Therefore, it seems that averages of both males and females (brain and body sizes) would be more appropriate comparisons in analyses where fossil hominins are included. If the authors choose not to do this, they should justify it.

In the source for our hominin data (Robson and Wood, 2008, Appendix I) body mass estimates are given for males and females of each hominin species. They note cases where sex determination was ambiguous, but this did not apply to any of the species in our analysis.

We added text to the subsection “Comparative data” explaining that female specific body mass estimates are available for extinct hominin species, and that we used female values mass because they are more tightly linked to ecological and life-history factors, while sexual selection can drive increases in male body mass unlinked to ecology, obscuring brain-body scaling relationships:

“Given that sex specific body mass estimates are available for ancient humans and extinct hominins [2], we used female values for body mass because female values are more tightly linked to ecological and life-history factors [39] and sexual selection can drive increases in male body mass unlinked to ecology, obscuring brain-body scaling relationships [40].”

2) Although there is some clarity about the phylogenetic relationships of extant primates, there is less consensus about the relationships of fossil hominins. The authors use a consensus tree of their own making, but it is not clear how the analysis would change if small details of this tree were shifted around. It would be extremely helpful if at least one alternative phylogeny was demonstrated to have little impact on the results. Otherwise, a minor shift in the phylogenetic status of a fossil taxon could invalidate the big take-home from this study.

Our consensus tree was actually obtained through a separate Bayesian phylogenetic analysis (Organ et al., 2011). We constructed an alternative “hominin phylogeny” (new terminology for “grafted phylogeny”) from a different consensus tree given in Organ et al., (2011). We describe this in the subsection “Comparative data”:

“To ensure that our results in this set of analyses were not dependent upon the topology of the hominin phylogeny, we repeated them using an “alternate hominin phylogeny,” constructed in a similar manner using another hominin tree from [13].”

We give additional details of how the hominin and alternate hominin phylogenies were constructed in subsection “S1: Data compilation”.

We re-ran the bayou analyses of ECV relative to body mass and of raw ECV with this phylogeny (and updated hominin data including *H. erectus*, see below). In addition to generating these new results, we also recognized and corrected a problem that was likely the source of some confusing results in the prior manuscript version. Previously, we found that the two MCMC chains that we ran for unweighted predictor OU model in the analysis ECV with no predictor identified very different patterns of evolution and differed significantly in likelihood. We determined that the chain with the very high likelihood was inferring an unrealistic pattern of evolution with a stationary variance approaching zero and many shifts with low support. We found that several of the new chains that we generated also suffered from this problem. To remedy this, we ran up to 6 chains for each model in each analysis of ECV, discarded the chains displaying this pattern of inference, and then selected the two chains with the highest mean likelihood to analyze further. We added text to the subsection “S3: Problems with MCMC convergence in bayou” to describe this issue and our solution.

When we analyzed ECV relative to body mass, we found that the pattern of shifts towards larger ECV was evident in every OU model we fit, regardless of the phylogeny we used. However, we found that the Brownian model was favored when we used the hominin phylogeny and an OU model was favored when we used the alternate hominin phylogeny. However, regardless of which mode of evolution is assumed to provide the best model fit, our results support patterns of accelerating evolution, as this result was found in the bayou results, BayesModelS results, and when fitting PGLS models of brain size deviation. We added text to the Discussion section to clarify this:

“We accounted for some phylogenetic uncertainty by repeating our analyses with an alternate phylogeny. The use of different phylogenies influenced outcomes of some statistical tests, as the Brownian model favored when we used the hominin phylogeny and OU model was favored when we used the alternate hominin phylogeny. However, we found that all of the OU models we fit inferred the same pattern of evolution towards larger ECV along the human lineage. The results of our outlier tests and PGLS model fitting – which assume a Brownian mode of evolution – also detected this pattern on different phylogenies. Collectively, these results indicate that our findings are likely to be robust to variations in assumed evolutionary relationships, and potentially to assumptions about the mode of evolution.”

When we analyzed ECV with no predictor variable, we found that Brownian models were favored over the OU models for both the hominin and alternate hominin phylogenies. Previously, OU models were favored. However, this was likely due to the high likelihood of one of the chains run for the unweighted predictor OU mode that inferred an unrealistic pattern of evolution. Because the Brownian model was favored, we added a new BayesModels analyses of ECV (using the tree block and data for humans and neanderthals); this new analysis did not identify humans as positive outliers for ECV. We interpret this result as “indicating that without correcting for body mass, the variance in ECV across primates is great enough to prevent humans’ brains from being detected as exceptionally large” (Results section).

We re-ran the BayesModels analyses of ECV relative to body mass while excluding all hominin data with this alternative phylogeny. We found that our results did not change qualitatively, and that changes in our numerical results were very minor. We summarize these findings in Figure 2—figure supplement 2 and Supplementary file 2.

Our results were also very similar when we fit PGLS models of brain size deviation (calculated in the new analysis) and divergence time, as we found evidence for directional and accelerating evolution towards larger brains since the *Pan-*hominin split. We summarize these results in Figure 3—figure supplement 1A. During our revisions, we determined that our description of these analyses could be improved, and that adding an additional PGLS model would make it easier for readers to interpret our results. We therefore added text in the Materials and methods section to provide a better description of the models we fit, and updated our findings, which were not qualitatively changed, in the Results section. We also decided that it is better to present the data alone in the figure. The model fits are very hard to interpret visually, as the scaling parameter δ changes the expected trait covariance among species in a way that is dependent upon the topology of the phylogeny. As an alternate means of displaying the accelerating trend identified by the model, we present Figure 3B showing the hominin tree would be transformed according to the best model, i.e. with longer branches indicating more change.

3) It would be very useful to include in the supplementary information a list of which specimens (museum numbers) of which fossil hominins went into the figure in each of the datasets. Sample sizes are so small for many of these, that a single cranium can make a big difference in this computation. The use and quality of different data sets has been a huge issue for this field, which has been bedeviled by small sample sizes, difficulties in dealing with intraspecific variation, and the use of estimated metrics (ECV, body mass) that may be derived in different ways esp. across fossil and extant species. Both ECV and body mass estimates are notoriously variable in hominin paleontology even for individual fossils, let alone in terms of a "species norm". This is a general issue that really needs to be discussed for the whole data set, but I would point to one particular point of concern which would be the source of the Homo sapiens data. From which population or populations, fossil or modern, does this derive? Implications esp. for body size could potentially be a factor in the different modeling results for H.s. vs. Neanderthals, just as the authors point out that unique traits of Gorilla may be a "byproduct" of body size change rather than brain change.

We added two new tables. The first (Supplementary file 2) lists the values for hominin ECV and the sample size associated with each value. The second (Supplementary file 3) lists the value and source of each human whole-brain trait (ECV, brain volume). We also include a description of each value, and clarify that our value for human ECV (from Robson and Wood, 2008) comes from 66 fossil specimens collected across a wide geographic area, and our values for human brain volume come from measurements of modern human brains.

We describe in subsection “S1: Data Compilation” that museum numbers for the hominin fossils are given in Robson and Wood, (2008). Given that the sample sizes are quite large, we believe it is most appropriate to refer readers to the original publication rather than re-list the >100 museum numbers in a table.

We also made added clarifications about our data and their sources in subsection “S1: Data Compilation”.

In the Discussion section, we added text discussing how data quality and intraspecific variation could affect our results. By using a data set of average hominin ECV/body mass, we avoid some of the statistical problems introduced by representing a species with a single fossil specimen. We also discuss how using an alternate phylogeny did not qualitatively change our results.

4) It is not clear why only a single non-African species – H. neanderthalensis – was included in this analysis, when there are many examples of ECV for H. erectus and H. heidelbergensis. If it is to restrict the analysis to evolutionary trajectories that could feasibly have led to modern humans, that is acceptable – but the authors should explicitly explain these decisions.

We did not include *H. heidelbergensis* in our analyses because its phylogenetic position is unresolved. We added a description of this (including a citation to a relevant study) in subsection “S1: Data Compilation”:

We agree that we should account for *H. erectus* in our analyses. The hominin phylogenies from Organ et al., (2011) treat *H. erectus* and *H. ergaster* as the same species. Therefore, we decided to redo our analyses after replacing data for *H. eragaster* with data for *H. erectus sensu lato,* which was calculated from fossils attributed to both *H. erectus* and *H. ergaster*. We describe this in subsection “S1: Data Compilation”.

We also update our findings (Results section and Figure 1, Figure 2 and Figure 3), which did not qualitatively change, for analyses of ECV.

5) Similarly, I do not understand why African H. heidelbergensis was not included in this analysis. Is this because its phylogenetic position is poorly-understood, and the authors were restricting themselves to African specimens where there is still debate over the number of Middle Pleistocene species of Homo? At the very least, this omission should be justified. It is a critical link between H. ergaster and H. sapiens.

As we state above, we did not include *H. heidelbergensis* because its phylogenetic position is not resolved, and was not included as a separate species by Organ et al., (2011, the source of our phylogeny).

Discussion Revisions:1) The current level of discussion of the results on medulla size evolution should be more explicit about their medulla analyses or just consider de-emphasizing these results. This is because predictions about why medulla volume might vary among primates is elusive. Either the authors should be more explicit about their medulla analyses or just consider de-emphasizing these results.

We added text in the Discussion section restating that previous analyses have found that medulla volume varies less between clades than the neocortical or cerebellar volume, and that our results were in line with these expectations.

We also clarified our expectations for medulla evolution in the Introduction.

We do believe that mentioning the medulla results is important because it further illustrates that brain components do not necessarily evolve in a linked manner, consistent with the mosaic theory of brain evolution.

2) Subsection “Endocranial volume (ECV)”: If the weak OU model provides a better fit and suggests that all primates experienced similar evolutionary trajectories, how does that mesh with the previous result that they show variation among taxa in evolutionary rate? More explanation and interpretation of this result seems warranted.

This discrepancy was due to the problems with the convergence of bayou models described above. After correcting this problem, we found that the Brownian motion provided a better fit, and the mismatch between results no longer exists.

3) Results of the non-human primates should be more thorough. Given the volume of interesting results, particularly in Table 2, spending more time probing and hypothesizing about individual changes is warranted. Analysis without interpretation and suggestion for future work leaves this section flat. There is some discussion of this here but given the support in the dataset, outlining some potential new specific hypotheses for how primate life history influences brain evolution would make this a much more powerful paper.

Thank you for this suggestion. We agree that more discussion of the life history implications of our findings and future directions for testing specific hypotheses is warranted. Thus, we made extensive additions (Supplementary Materials) to discuss how our results for non-human primates broadly relate to primate life history. We also added some description of our finding that several non-human primates exhibit a reduction in the size of a brain component. We discuss this result in the context of previous research and make suggestions for future work. Finally, we elaborated on how future research could test more specific ecological hypotheses.

Although further analyses are needed to determine the causes and consequences of the changes we observed, these analyses are outside the scope of this paper. We hope these additions – including new citations – will help to motivate future research along these lines.

4) There are of course many different ways to characterize the "size" of the human brain: absolute, relative to body, relative to other brain part, relative to range size, relative to phylogeny, etc. As it stands the discussion seems to imply that our task is simply to select and apply the one correct way, but I suspect/hope the authors would agree that the most appropriate method depends on the question being asked. For example, many studies of comparative cognition suggest we should actually focus on absolute rather than relative brain size whereas the choice to control for body size or phylogeny is also related to certain assumptions and interests. These authors would be very well suited to provide a definitive discussion of this issue that would be a useful touchstone for the field, and would also more clearly articulate the specifically evolutionary (e.g. as opposed to functional) questions their method is optimized to address. Some of this is indirectly reflected in the discussion of the unexpected results for the chimpanzee sub-species and it would be nice to see a more comprehensive treatment.

We added a discussion of the interpretation of absolute brain volume vs. brain volume relative to body mass in the Introduction.